# Effects of Group Music Intervention on Depression for Elderly People in Nursing Homes

**DOI:** 10.3390/ijerph19159291

**Published:** 2022-07-29

**Authors:** Ai-Ling Yu, Shu-Fen Lo, Pei-Yu Chen, Shiou-Fang Lu

**Affiliations:** 1Department of Nursing, St. Mary’s Junior College of Medicine, Nursing and Management, Sanshing Township, Yilan 26647, Taiwan; ayling@smc.edu.tw; 2Department of Nursing, Tzu Chi University, Hualien 97074, Taiwan; shufen5712@gms.tcu.edu.tw; 3Center for Health and Welfare Data Science, Tzu Chi University, Hualien 97074, Taiwan; swf0409@gms.tcu.edu.tw

**Keywords:** depression, elderly, live music performance, group music intervention, nursing home

## Abstract

Depression is the most common mental problem among the elderly, especially in long-term care facilities. The purpose of the present study was to examine the effects of group music intervention on depression for elderly people in nursing homes. Methods: A randomized control trial consisting of sixty-three elderly participants randomly and blindly assigned to a music group or control group was utilized. The music group received 20 sessions of group music intervention (two 30-min sessions per week for 10 weeks), and the control group received usual care with no music intervention. The Geriatric Depression Scale—Short Form (GDS-SF) and salivary cortisol at baseline, 5 weeks, and 10 weeks were collected for analysis. Results of the GEEs (generalized estimating equations) analysis indicated that after 20 sessions for 10 weeks of group music intervention, the groups showed a statistically significant difference in depression at 5 weeks and 10 weeks. There was no significant difference in the salivary cortisol concentration between the two groups. The results show that the group music intervention may effectively reduce the depression scores for elderly people in nursing homes. Conclusion: The group music intervention has positive effects on depression.

## 1. Introduction

Global population aging and depression are becoming a significant issue, as the number of people aged over 80 is expected to triple from 143 million in 2019 to 426 million in 2050 [1]. Depression in the elderly is common, and associated with major functional impacts, impaired quality of life, and a significant risk of suicide. Several studies have reported a marked increase in the prevalence of depression and depressive syndromes in elderly residents of nursing homes, as much as three to four times higher than in the community [2,3].

A meta-analysis by Barua et al. [4] of 74 studies, including 487,275 elderly individuals, found that the worldwide prevalence rate of depressive disorders to be between 4.7% and 16%. Furthermore, 74% of post stroke older adults had depressive symptoms [5].

The prevalence of depression increases with advancing age. In addition, 21.1% to 50% of elderly residents in a nursing home have clinically significant depressive symptoms, even in the absence of major depression [6]. Significant risk factors that were found to be associated with depression were the length of stay for more than 2 years, known history of depression, pain, and none or lack of social contact [6,7].

Depression shows a significant association with the age of the elderly [7,8]. The highest risk factors were lack of social engagement, low family support, chronic disease, and disturbed sleep [3,8]. Multiple studies corroborate that the prevalence rates for depression were high among elderly people in long-term facilities or nursing homes. More attention is needed to elucidate the psychosocial health of elderly residents in long-term facilities. While medications are effective in controlling depression in elderly people, various facility, provider, patient factors, safety, and drug interactions must be carefully controlled [9].

Music therapy as “the clinical and evidence-based use of music intervention to accomplish individualized goals within a therapeutic relationship by a credentialed professional who has completed an approved music therapy program” may prove to be an effective non-pharmaceutical treatment modality [10].

A meta-analysis by de Witte et al. [11] of 104 RCT music interventions and 9617 participants were found to have a significant effect on the reduction in both physiological and psychological stress. Chronic stress is associated with depression. High cortisol concentration, a biomarker of an activated stress response, has been found in depressed patients [12]. Several music intervention studies used cortisol as an indicator of physiological outcomes for music intervention [13,14].

Several studies have reported that music therapy is an effective intervention for depression [13,15,16,17,18], improving cognitive function [19,20,21] and the behavioral and psychological symptoms of dementia (BPSD) [22,23,24,25]. Most music interventions use group music therapy approaches [13,17,18,26,27], while some studies used individual music therapy [22,25].

There are various types of music therapy. Music-based interventions can be divided into two categories: passive and active. In active music therapy, the therapist and participants actively create music, sing, dance, etc. In passive music therapy, the participants mainly listen to the music and relax. Most music therapy is carried out twice a week [13,24,28] and commonly lasts for 10 weeks [19,28,29]. While the forms of music intervention are diverse, few studies have evaluated the effects of live music performances of passive music therapy on depression for elderly people in nursing homes. Therefore, the aim of this study was to examine the effects of group music intervention on depression for elderly people in nursing homes. This study specifically used passive music therapy with live music performances. The hypothesis of this study was that music intervention can improve depression among elderly people in nursing homes.

## 2. Materials and Methods

### 2.1. Research Design

This study was designed and conducted as a randomized controlled trial to evaluate the effects of group music with live music performance intervention on depression among elderly people in nursing homes. Participants were randomly assigned to the music group or the control group. Randomization was conducted by a person independent of the researcher, using a computer-generated list of random numbers.

### 2.2. Participants and Setting

The study population consisted of 238 elderly people from two nursing homes. The inclusion criteria for participation were as follows: (1) an age of 65 or above; (2) living in the nursing home for at least more than 1 month; and (3) assessment by the Short Portable Mental Status Questionnaire (SPMSQ) presented normal or mild cognitive impairment. SPMSQ is used as a screening tool to ensure that they can answer depression questions as well as basic information.

The exclusion criteria were (1) unconscious or severe hearing impairment; (2) declined to participate; and (3) unable to participate in an intervention for at least 30 min. Participants were randomly divided into two groups by a computer-generated random sequence.

The statistical computer program GPOWER was used for a priori calculation of the required sample size. The calculation was based on a two-tailed test. A conventional alpha (0.05), conventional power (0.8), and an effect size of 0.75 required a sample of 58 patients. Accounting for a normal dropout rate, the sample size for each group was 35. Patients who met the study criteria were approached and 70 patients gave their written informed consent to participate. They were also informed that they retained the right to withdraw from the study at any time.

The study was carried out in two nursing homes located in Yilan, Taiwan. The service targets of the two nursing homes are the elderly who are stroked, half-paralyzed or paralyzed, and unable to take care of themselves due to chronic diseases. They provide residents with nursing care, daily care, replacement of tube irrigation, body cleaning, etc.

### 2.3. Intervention and Procedure

Both the music group and control group received their prescribed medication. Usual care in the nursing home included 24 h care with activities of daily living, basic nursing care, meal provision, and social activities (e.g., TV watching, family visiting, and occasional parties for special events). The participants in the music group also received the same care as the control group. The music group attended 30-min group music therapy sessions twice a week for 10 weeks. The music therapy intervention was led by a nurse who is experienced in conducting music therapy and also has worked in the nursing home for many years.

Previous research showed that personal preference music was more efficient [30,31]. Before beginning the group music intervention, the researchers collected data on participants’ music preferences and prior music performance experiences. It was found that elderly people generally preferred to listen to Taiwanese and Chinese songs from the 1950s to 1960s. Various physical and mental problems affect elderly people that may affect their attendances to the music therapy; therefore, the music therapy was designed as 30-min sessions and conducted twice a week at 2:00 pm. The group music comprised 20 sessions of passive music interventions. This included music background and instrument introduction for 5 min, listening to live music performances for 15 min, and sharing listening experiences for 10 min. The instruments used in the solo performances included a saxophone, piano, violin, and flute. The music performer was a volunteer with a medical background.

### 2.4. Ethical Consideration

This study was approved by the Research Ethics Committee of the Hospital (approval registration number IRB106-97-B). We provided eligible nursing home residents and their authorized representatives with a detailed explanation of the study and invited them to participate. Informed consent was obtained from each participant and the participants’ families who agreed to join the study.

### 2.5. Measures

#### 2.5.1. Demographics

Background information included demographic characteristics (gender, age, marital status, education) and disease characteristics (number and type of chronic diseases, length of stay in the institution).

#### 2.5.2. Geriatric Depression Scale—Short Form

Depression status was the primary outcome. The Chinese version of the Geriatric Depression Scale—Short Form (GDS-SF) to assess depression status was employed [32]. The GDS-SF consists of 15 yes/no questions with a score ranging from 0 to 15. The higher the score, the more severe the patient’s depressive condition is. The Cronbach’s alpha value of the GDS-SF was found to be 0.875 in the present study.

#### 2.5.3. Salivary Cortiso

Salivary cortisol as a biological marker of change in the depression level in response to the intervention was utilized. The cortisol levels were quantified using a Cortisol ELISA kit (Cayman Chemical Number 500360). Before collecting the saliva samples, the participants were asked to rinse their mouth with water for 6–10 min. In order to avoid interrupting the patients’ daily schedule and causing confounding effects from their physiological cycle, the saliva collection was conducted at 8:30 a.m. Furthermore, studies found that the morning cortical concentration is correlated with depression [33,34]. Therefore, one hour after breakfast, a 5 mL saliva sample was collected from each participant. These samples were first stored in a −20 °C freezer and then sent on dry ice to the biochemical laboratory for analyses.

### 2.6. Data Collection and Data Analysis

#### 2.6.1. Data Collection

Demographic information was provided by the participants or institutions. GDS-SF and salivary cortisol were assessed 3 times: baseline before the intervention (Time 1); the 10th session of the intervention (Time 2); and the 20th session of the intervention (Time 3) by two blinded assessors. All the participants’ depression scale and salivary cortisol data were collected for analysis. Assessors who did the GDS-SF and salivary cortisol assessment for participants were blinded. Assessors did not know the participant’s respective group.

#### 2.6.2. Data Analysis

Data were analyzed using Statistical Package for the Social Sciences (SPSS) version 21.0 for Windows (SPSS, Inc., Chicago, IL, USA). Homogeneity test for the general characteristics of two groups consisted of the χ^2^ test and Fisher’s exact test. Inferential statistics were conducted using generalized estimating equations (GEEs) to estimate the repeated measures and analyze the impact of group music therapy on depression and salivary cortisol. For each variable, a GEE model including the covariates of group, time, group-by-time interaction, and baseline was constructed for evaluating the effects of the group and group-by-time interactions. If the interaction effect occurred, an ANCOVA model including the covariates of group and baseline was conducted at each time point to evaluate the group effects. The statistical significance was accepted as *p* < 0.05 for this study.

## 3. Results

### 3.1. Participant Flow

Figure 1 depicts the CONSORT diagram for participants. A total of 238 participants were screened during the study. Among these participants, 173 did not meet the inclusion criteria and 2 declined to participate. The remaining 63 residents were randomly allocated into two groups; 32 were allocated to the music group and 31 were allocated to the control group. During the ten-week intervention period, 58 participants (92%) completed the study, two participants withdrew from the music group due to lack of interest and hospitalization, and three participants withdrew from the control group due to physical illness and discharge. A total of twenty group music therapy sessions were conducted with an average attendance of 18 sessions, accounting for the attendance rate of 90%.

### 3.2. Demographic Information

The demographic and clinical characteristics of the participants at baseline assessment can be seen in Table 1. Except for gender (*p* > 0.05), no differences were found at baseline between the music group and control group for any of the outcome variables.

The mean age of the participants was 80 (SD = 7.35), including 35 females (55.6%) and 28 males (44.4%). A total of 46 were widowed or divorced (76%). Most of them were illiterate (47.6%). The mean length of stay years was 2.83 years (SD = 3.22).

No significant differences in the GDS-SF and salivary cortisol levels were exhibited at the baseline assessment (Table 2). The music group and control group had baseline average GDS-SF scores of 8.65 (±3.38) and 10.16 (±3.91), showing that participants in both groups exhibited a certain depression status.

### 3.3. Result on Outcome Measures

The mean GDS-SF scores for the music group gradually declined from 8.65 (±3.38) at baseline to 3.96 (±1.54) and 2.96 (±1.29) at Time 2 and 3, respectively. The mean GDS-SF scores for the control group varied only slightly from baseline 10.16 (±2.91) to 10.39 (±2.87) and 10.17 (±2.77) at the same time points, respectively (Table 2). The difference in baseline depression scores and salivary cortisol level between the music group and control group was not significant (Table 2). Inferential statistical analysis using GEE was then conducted. After adjusting for time, group, and gender effects, the differences in mean GDS-SF scores for the time, group, and time-by-group interaction in the music group and control group was analyzed. The time-by-group interaction exhibited a significant difference for depression (GDS-SF) (Wald χ^2^ = −0.78; *p* < 0.0001). After adjusting for time, group, and gender effects, differences in the mean salivary cortisol for time, group, and time-by-group interaction in the music group and control group were analyzed. There was no significant difference in change in salivary cortisol between the music group and control group (Wald χ^2^ = 100.12; *p* = 0.686; Table 3).

For the ANCOVA result (controlling for baseline GDS and gender), the time-by-group interaction exhibited significant differences for the GDS-SF at the 5th week (F = 250.48, *p* < 0.0001) and the 10th week (F = 232.32, *p* < 0.0001) scores. Therefore, the main effects on depression indicated a significant group effect over time. The effect size was 2.75 at the 5th week and 2.51 at the 10th week, indicating a very large effect size (Table 4).

## 4. Discussion

The effectiveness of group music intervention on depression for elderly people in a nursing home was examined. The result of this study showed that after 20 sessions for 10 weeks of live music performance group intervention, the groups showed a statistically significant difference in depression at 5 weeks and 10 weeks. There was no significant difference in salivary cortisol concentration between the two groups.

### 4.1. Effects on Depression

In this study, receiving the group music intervention was sufficient to reduce the depression for elderly people in nursing homes. These results corroborate the studies by Chu et al. [10], Perez-Ros [17,24], and Raglio et al. [24].

This study involved a 30-min music intervention twice a week for 10 weeks, which was the same as used in the study conducted by Raglio et al. [24]. In addition, this study involved passive music therapy, which was the same as Perez-Ros’s study [17]. However, the unique aspect about the present study was that the participants were listening to a live music performance as the intervention. The musicians were volunteers with medical backgrounds. Chu [10] et al. and Raglio [24] used both active and passive music interventions.

A total of 10 h of live music intervention was employed in this study, which differs from Raglio’s 40-h intervention [24]. The present study suggests better economic efficiency to attain the goal of depression reduction within fewer hours.

This study is the first randomized clinical trial in Taiwan to apply group music intervention in the form of listening to live performances on the elderly in long-term care institutions. These songs were chosen and performed in this study. When the participants heard the music they knew well and loved, they were observed to smile and appeared happy. Wilkines [30] found that listening to one’s favorite song alters the connectivity between auditory brain areas and the hippocampus.

### 4.2. Effects on Salivary Cortisol

The salivary cortisol biological marker of change in depression level in response to the intervention after 20 sessions for 10 weeks of group music intervention showed no significant difference in salivary cortisol concentration between the two groups.

A two-year follow-up study did not support the hypothesis that a high salivary cortisol concentration is a risk factor of depression [12].

Similar to the study by Chu [10], the present results indicate no significant changes in salivary cortisol concentration between the two groups of the music intervention. One important difference from the de la Rubia [28] study, was that the cortisol level was collected in the morning while the study by Chu [13] collected the salivary samples at noon. Both Chu [10] and de la Rubia [14] collected samples 15 min post music therapy. In the present study, the group music intervention took place at 2:00 p.m. The saliva samples were collected at 8:30–11:00 a.m. the next day after the 5-week and 10-week music intervention, respectively. The timing of collection may affect the results of the salivary cortisol analysis.

### 4.3. Strengths and Limitations

This study benefits from some strengths and suffers various limitations. The strengths of the present study included the use of the subjective and objective outcomes measures and a randomized sampling technique. However, this study was limited due to the single-blind design. Only the data collectors did not know the participants’ group while the participants knew their assigned groups. Moreover, the measurement time was only set at before, during, and after the music intervention. This design also lacks long-term follow-up of the group music intervention effect. Resource limitations that restricted the study sites to two nursing homes for elderly people may limit the generalizability of the study results. We recommend that future studies use larger samples, have more outcomes, and more long-term follow-up measures to increase the empirical understanding of the impact of music therapy.

## 5. Conclusions

The present study indicates 20 sessions for 10 weeks of group music intervention had a positive effect on reducing depression among elderly people in a nursing home. The research results confirmed the effects of group music therapy on ameliorating depression in nursing home residents.

These points can be suggested as the results of this study: Since nurses interact closely with nursing home residents, they should be educated and trained in group music therapy to better help residents ameliorate depression. In addition, it is suggested that group music therapy should be used as routine care in nursing homes. Group music therapy is an economical and easily administered method for ameliorating depression among elderly people in nursing homes.

## Figures and Tables

**Figure 1 ijerph-19-09291-f001:**
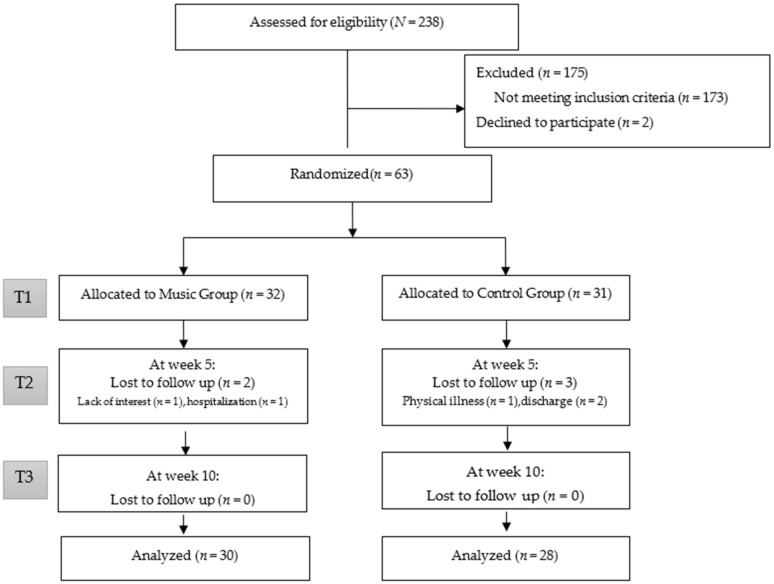
CONSORT diagram for the participants in the study.

**Table 1 ijerph-19-09291-t001:** Homogeneity of the general and clinical characteristics and variables among groups.

Characteristic		Music Group (*n* = 32)	Control Group (*n* = 31)	*p* Value
Gender	Male	10 (31.3%)	18 (58.1%)	<0.05
	Female	22 (68.7%)	13 (41.9%)	
Marital status	Married	5 (15.6%)	12 (38.7%)	0.07
	Widowed/divorced	27 (84.4%)	19 (61.3%)	
Religion	No	1 (3.1%)	2 (6.5%)	0.62
	Taoism	28 (87.5%)	24 (77.4%)	
	Buddhism	3 (9.4%)	4 (12.9%)	
	Christian	0 (0%)	1 (3.2%)	
Education	Illiterate	15 (46.9%)	15 (48.4%)	0.51
	literate	2 (6.3%)	1 (3.2%)	
	Primary	12 (37.5%)	7 (22.6%)	
	junior	1 (3.1%)	2 (6.5%)	
	High school	2 (6.3%)	5 (16.1%)	
	College	0 (0%)	1 (3.2%)	
Health status	None	1 (3.1%)	0 (0%)	0.65
	One item	10 (31.3%)	9 (29.0%)	
	Two item	10 (31.3%)	12 (38.7%)	
	Three item	11 (34.4%)	9 (29.0%)	
	>Four item	0 (0%)	1 (3.2%)	
Mean age (SD)		80.30 ± 6.88	79.03 ± 7.71	0.19
Mean length of stay years (SD)		3.16 ± 3.26	2.49 ± 3.20	0.40
GDS-SF(baseline)		8.65 ± 3.38	10.16 ± 2.91	0.054
Salivary cortisol(pg/mL) (baseline)		585.13 ± 573.74	567.66 ± 541.49	0.317

Values are the mean ± standard deviation or *n* (%); tested by χ^2^ tests and Fisher’s exact tests.

**Table 2 ijerph-19-09291-t002:** Mean values of depression and salivary cortisol at different timepoints.

Variable	Time 1 (*N* = 63)		Time 2 (*N* = 58)		Time 3 (*N* = 58)	
	*M (SD)*		*M (SD)*		*M (SD)*	
	Music Group	Control Group	Music Group	Control Group	Music Group	Control Group
GDS-SF	8.65 (3.38)	10.16 (2.91)	3.96 (1.54)	10.39 (2.87)	2.96 (1.29)	10.17 (2.77)
Salivary cortisol	585.13 (573.74)	567.66 (541.49)	765.16 (746.18)	866.67 (728.43)	645.61 (668.74)	647.00 (759.34)

GDS-SF = Geriatric Depression Scale—Short Form 15; Values are the mean + standard deviation.

**Table 3 ijerph-19-09291-t003:** GEE model results for depression (GDS-SF) and salivary cortisol by group and time interaction (*N* = 58).

	Depression(GDS-SF)			Salivary Cortisol		
Effect	Wald χ^2^	df	*p*	Wald χ^2^	df	*p*
Intercept	4.43	1	<0.001	680.92	1	0.051
Group	−4.84	1	<0.0001	−272.21	1	0.511
Time	−0.21	1	0.092	−219.67	1	0.23
Group × Time	−0.78	1	<0.0001	100.12	1	0.686
Baseline	−0.45	1	<0.0001	−0.86	1	<0.0001

GDS-SF = Geriatric Depression Scale—Short Form 15; Intercept = intercept term; Group = treatment group; Time = time points; Group × Time = interaction term between treatment group and time points; Baseline = GDS-SF and salivary cortisol measures at baseline (T1); Wald χ^2^ = test of hypotheses on parameters estimated by maximum likelihood; df = degrees of freedom.

**Table 4 ijerph-19-09291-t004:** Effect of music intervention on depression at different timepoints (*N* = 58).

Variable	Type III Sum of Squares	df	Mean Squares	*F*	*p* Value	Effect Size Cohen *d*
Effect on depression (GDS-SF)						
5th weekly						
Intercept	1.87	1	1.87	1.21	0.274	
GDS (baseline)	83.54	1	83.54	54.32	<0.0001	
Gender	2.16	1	2.16	1.41	0.24	
Group	385.16	1	385.16	250.48	<0.0001	2.75
Error	83.03	54	1.53			
10th weekly						
Intercept	5.29	1	5.29	2.29	0.136	
GDS (baseline)	146.42	1	146.42	63.41	<0.0001	
Gender	2.28	1	2.28	0.99	0.324	
Group	536.43	1	536.43	232.32	<0.0001	2.51
Error	124.68	54	2.3			

ANCOVA (analysis of covariance); GDS-SF = Geriatric Depression Scale—Short Form 15.

## Data Availability

Not applicable.

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
