# Peer review of "Effects of Group Music Intervention on Depression for Elderly People in Nursing Homes"

_ijerph, 2022, doi:10.3390/ijerph19159291_

Round 1
Reviewer 1 Report
I thought that it was interesting. I recommend that the authors revise the manuscript beyond my suggestions to ensure proper grammar, consistent punctuation, consistent use of capitalization, and sentence structure.
Specific comments.
1. Introduction
The introduction could be revised to explain the main purpose of this study more logically. I suggest clarifying in paragraph 6 how live music interventions differ from other music therapies.
The third sentence of paragraph 1 suggests adding references.
2.2. Participants and setting
I am very confused by the inconsistent description in the first and fourth paragraphs of part 2.2. I do not understand whether the subjects of this study come from one nursing home or two nursing homes.
2.3. Intervention and Procedure
The twice interventions per week in this study did not specify the specific intervention time points, and different intervention time points may form confounding factors that interfere with the results of the study. For example, depression has a marked circadian rhythm and therefore the time of day at which it is reported will affect its level.
Before starting the group music intervention, the researchers collected data on participants' musical preferences and previous music performance experiences, a good move to control for confounding variables. However, prior music therapy was not considered in the inclusion and exclusion criteria for subjects in this study.
3.3. Result on Outcome Measures
In section 2.5.3, the author showed that salivary cortisol was a biological marker of depression level changes after intervention. However, the results of this study showed no statistically significant difference in salivary cortisol levels between the music group and the control group.
4.Limitations
All participants selected in this study were only from one or two nursing home, so there may be selection bias in choosing participants of the study.
Author Response
Thank you for your encouragement and giving me the opportunity for revision. We have revised the manuscript according to the reviewers’ comments, and responses to the reviewers’ comments are provided. Thank you for the valuable advice and thanks again for your consideration.
Best regards

Reviewer 2 Report
SCOPE: the manuscript is in line with the thematic scope of the International Journal of Environmental Research and Public Health.
TITLE: it is more informative and it specifies the type of research.
ABSTRACT: the length and quality are correct.
KEYWORDS: number of keywords is acceptable.
INTRODUCTION: the introduction quite briefly describes the research issues, what gap does this research fill in the current knowledge?
METHOD: what hypotheses were tested? How were the respondents selected to ensure that the sample was representative? What kind of draw was used? What sampling frames were used - was it a complete list of nursing home residents?
RESULTS: the results of the focus study are properly presented and described. The presented data are sufficient to draw conclusions.
DISSCUSSION: this section is limited in quantity and quality. What, according to the authors, does this specific single RCT prove?
CONCLUSIONS: there is one cognitive conclusion and one postulative one (it is not a critical remark).
REFERENCES: the number of bibliography items is average and is represented by items more than 10 years old.
Author Response

(The authors gave the same response as above.)

Reviewer 3 Report
Thanks for the opportunity to review this manuscript. In this paper, the authors describe the methods and results of a randomized clinical trial of a group music intervention and its effects of depressive symptoms and salivary cortisol. The strengths of this paper are: 1) a relatively large sample of older adults recruited from nursing homes; 2) a single blinded nature of the study; and 3) details explanations of the depression prevalence among older adults living with nursing home. The main weaknesses are: 1) very limited introduction which does not quite set up the study, intervention, and study outcomes; 2) lack of details in the methods; 3) discussion section is not very well developed and does not situate this study in the field. I also provide minor edits throughout.
1. Title: the title does not reflect and match how the intervention is described in the rest of the manuscript.
1.1. Since the sample included those with mild cognitive impairment, the title should reflect that population as well.
2. Abstract
2.1. Abstract says group music intervention which does not match the title.
2.2. It says that there were statistically significant differences for depression among groups, but in what direction?
2.3. Results should include data analysis methods and statistics.
2.4. Unclear how depression was measured in the abstract.
2.5. Last sentence states that there was an increase in interpersonal interactions, but this was not the outcome of the study. This needs to be changed.
3. Introduction
3.1. Missing references: 1) first paragraph after “higher than in the community” ; 2) after “non or lack of social contact”
3.2. Second and 3rd paragraphs feel more like a listing of different studies, rather than a summary of previous literature. There needs to be more cohesion between the studies mentioned.
3.3. While there is a lot in the Introduction on depression, there is no mention of cortisol and how cortisol can be used to estimate depressive symptoms. It comes as a surprise that cortisol was used to measure depression, while it is customarily thought to measure stress response.
3.4. Unclear what is meant by ”needed to elucidate the psychosocial needs of elderly residents” – how does this study answer this gap?
3.5. The authors provide the definition of music therapy, but it is unclear why they do so. What they provide is not considered music therapy, so it is unclear why the definition is provided. It can be helpful to provide it, but perhaps in a discussion of music therapy vs music intervention and then describe what the intervention in this study is. Introduction can be much improved by having one paragraph focus on differences between the two methods and connecting that discussion to the intervention at hand.
3.6. Statement on page 2, 2nd paragraph is not quite correct. Not all citations 15-17 are systematic reviews.
3.7. It is unclear what the authors mean by “the intervention method divide used an individual music therapy…”
4. Methods
4.1. There are lot of details missing from the methods section which I outline below.
4.1.1.At first, I thought that 238 was a number of participants in the study. However, it seems that 238 is the number of current residents in two nursing homes. This needs to be clarified.
4.1.2.Need more details in the inclusion/exclusion criteria. For example, how was severe hearing impairment defined? What is meant by undesirable and unwilling elders?
4.1.3.Is “retarded” supposed to be “retained”?
4.1.4.More details need to be provided about the consent process, especially for those who have mild cognitive impairment.
4.1.5.More details need to be added to the description of the intervention. For example, who led these sessions? How was music chosen for the group? Who were the musicians?
4.1.6.Measures do not contain references to the instruments used.
4.1.7.More information needs to be included on the analysis of cortisol. Why was AM time selected for cortisol collection?
4.1.8.How was literacy assessed? The authors should provide details on how all variables (including covariates) were assessed.
4.1.9.When were these data collected?
4.2. Results
4.2.1.Figure 1 has some text cut off and doesn’t match the number presented in the text.
4.2.2.Figure 1. Also the reasons for dropping out were not listed.
4.2.3.Did the authors track attendance at sessions? It is hard to comment on the efficacy if the authors do not provide rates of attendance.
4.2.4.Minor: what is the acronym MIG stand for?
4.2.5.AVCOMA should be ANCOVA, I think.
4.2.6.Although not statistically significant, the intervention group had lower GDS scores, which should be addressed at some point in the Discussion.
4.2.7.How many of these individuals were diagnosed with mild cognitive impairment? These data are not provided.
4.2.8.Were individuals with cognitive impairment treated and analyzed the same way?
4.2.9.The authors were not able to recruit the targeted sample. This point needs to be addressed in the paper including the reason(s).
4.3. Discussion
4.3.1.Overall, the discussion is not organized around main results and does not place this study in the field.
4.3.2.Page 7 4th paragraph, how did the authors know that participants spent 10 hours in the intervention? This relates to my point above re: attendance tracking.
4.3.3.The authors suggest the based on this study findings there should be practice change. I think the authors should be careful in discussing direct positive implications of this work, rather they should focus their discussion on how a program like this can be implemented in nursing homes or what kinds of barriers can be faced by staff/residents (drawn from related literature).
4.3.4.Discussion on cortisol needs to be expanded. How does the timing of cortisol collection influence the outcome exactly?
4.3.5.Some of the information is repeated directly from the results section (for example, like most of paragraph 3 on page 8)
4.3.6.The strengths and limitations section is rather limited as well.
4.3.7.Implications and future research directions should be included at the end of the Discussion.
4.3.8.The authors mention social stimulation as a potential outcome of this study, however, they do not include it as an outcome in the actual study. This statement should be removed or more details should be provided on how social interactions were measured.
Author Response

(The authors gave the same response as above.)

Round 2
Reviewer 1 Report
I don't have any questions. Accept.
Reviewer 2 Report
Necessary changes have been made - I accept it.